# Computational remodeling of an enzyme conformational landscape for altered substrate selectivity

Antony D. St-Jacques [1,2], Joshua M. Rodriguez[3], Matthew G. Eason[1,2], Scott M. Foster[1,2], Safwat T. Khan[1,2], Adam M. Damry[1,2], Natalie K. Goto[1,2], Michael C. Thompson [3] & Roberto A. Chica [1,2] ✉

Structural plasticity of enzymes dictates their function. Yet, our ability to rationally remodel enzyme conformational landscapes to tailor catalytic properties remains limited. Here, we report a computational procedure for tuning conformational landscapes that is based on multistate design of hinge-mediated domain motions. Using this method, we redesign the conformational landscape of a natural aminotransferase to preferentially stabilize a less populated but reactive conformation and thereby increase catalytic efficiency with a non-native substrate, resulting in altered substrate selectivity. Steady-state kinetics of designed variants reveals activity increases with the non-native substrate of approximately 100-fold and selectivity switches of up to 1900-fold. Structural analyses by room-temperature X-ray crystallography and multitemperature nuclear magnetic resonance spectroscopy confirm that conformational equilibria favor the target conformation. Our computational approach opens the door to targeted alterations of conformational states and equilibria, which should facilitate the design of biocatalysts with customized activity and selectivity.

Enzymes are flexible macromolecules that sample multiple structural states, described by a conformational energy landscape[1]. It is the relative stability of these conformational states, and the ability of enzymes to transition between them, that ultimately dictates enzymatic function[2–4]. Analyses of directed evolution trajectories have shown that evolution can reshape enzyme conformational landscapes by enriching catalytically productive states and depopulating non-productive ones[5–7], leading to enhanced catalytic activity. Similar mechanisms contribute to the evolution of substrate selectivity, where transient conformations responsible for activity on non-cognate substrates become enriched[8–10]. Thus, it should be possible to harness the pre-existing conformational plasticity of enzymes to tailor their catalytic properties by reshaping their conformational landscapes. However, predicting the effect of mutations on these energy landscapes

remains challenging, and current computational enzyme design protocols, which focus on a single structural state, are poorly optimized for this task. New design methodologies are therefore required for the targeted alteration of subtle conformational states and equilibria, which would in turn facilitate the design of biocatalysts with customized activity and selectivity.

Here, we report a computational procedure for rationally tuning enzyme conformational landscapes that is based on multistate computational protein design, a methodology that allows protein sequences to be optimized on multiple structural states[11]. As a case study, we remodel the conformational landscape of aspartate aminotransferase, an enzyme that switches between open and closed conformations via hinge movement, which involves the rotation of a protein domain relative to another around an axis between two planes.

[1]Department of Chemistry and Biomolecular Sciences, University of Ottawa, Ottawa, ON K1N 6N5, Canada. [2]Center for Catalysis Research and Innovation, University of Ottawa, Ottawa, ON K1N 6N5, Canada. [3]Department of Chemistry and Biochemistry, University of California, Merced, Merced, CA 95343, USA. ✉e-mail: rchica@uottawa.ca

Using our approach, we enrich the less populated but catalytically active closed conformation in order to increase catalytic efficiency ($k_{cat}/K_M$) with the non-native substrate L-phenylalanine, leading to altered substrate selectivity. Steady-state kinetics reveal $k_{cat}/K_M$ increases of up to 100-fold towards this aromatic amino acid, resulting in a selectivity switch of up to 1900-fold, and structural analyses by room-temperature X-ray crystallography and multitemperature nuclear magnetic resonance (NMR) spectroscopy confirm that the conformational landscape is remodeled to favor the target state. Our methodology for altering conformational equilibria should be applicable to many enzymes and proteins that undergo hinge-mediated domain motions.

## Results

### Computational remodeling of conformational landscape

*E. coli* aspartate aminotransferase (AAT) is a pyridoxal phosphate (PLP)-dependent enzyme that catalyses the reversible transamination of L-aspartate with α-ketoglutarate, yielding oxaloacetate and L-glutamate (Supplementary Fig. 1). During its catalytic cycle, AAT undergoes hinge movement to switch between an open conformation, in the ligand-free form, and a closed conformation upon association with substrates or inhibitors (Fig. 1a,b)[12]. Previously, AAT was redesigned to change its substrate specificity to allow transamination of aromatic amino acids[13]. To do so, six of the 19 residues that are strictly conserved in AAT enzymes were replaced by those found at corresponding positions in the homologous *E. coli* tyrosine aminotransferase. These residues were selected because they line the substrate channel leading to the catalytic pocket or are located near the cofactor phosphate moiety. This process yielded a hexamutant (HEX) of AAT that is approximately two orders of magnitude more catalytically efficient than the wild type (WT) with the aromatic amino acid L-phenylalanine (Table 1), and similarly efficient for transamination of L-aspartate. Unexpectedly, it was found that unlike WT, HEX was closed in its ligand-free form (Fig. 1c)[14], demonstrating that those six mutations shifted its conformational equilibrium to favor the closed state (Fig. 1d), which is the active conformation of the enzyme[15]. This equilibrium shift was accompanied by enhanced L-phenylalanine transamination activity resulting from a larger increase in affinity for this non-native substrate than for the native L-aspartate substrate[13]. Since domain closure in AAT is driven by the excess binding energy of the native substrate L-aspartate, which forms strong polar interactions with two active-site arginines[15,16], the observed equilibrium shift towards the closed state in the absence of ligand was proposed to facilitate access to this active conformation for non-native substrates whose binding may not provide sufficient energy to induce domain closure due to their lower structural complementarity to the active site[14]. Based on these observations, we postulated that we could rationally remodel the conformational landscape of AAT by using multistate computational protein design[11] to identify novel mutation combinations that can preferentially stabilize the closed conformation over the open conformation, and in doing so, increase catalytic efficiency and selectivity for L-phenylalanine.

To test this hypothesis, we implemented a computational strategy (Fig. 2) that proceeds in five steps: (1) identification of hinge-bending residues involved in transition between open and closed conformations; (2) generation of structural ensembles approximating backbone flexibility to model open and closed conformational states; (3) optimization of side-chain rotamers for all allowed amino-acid combinations at key hinge-bending residues and neighboring positions, on each ensemble; (4) calculation of energy differences between open and closed states to predict preferred conformation, and (5) combinatorial library design using computed energy differences to select mutant sequences for experimental testing.

To identify hinge-bending residues for the open/closed conformational transition, crystal structures of WT AAT in its open and closed forms (PDB ID: 1ARS and 1ART, respectively)[12] were used as input for hinge movement analysis with DynDom, a program that identifies domains, hinge axes, and hinge bending residues in proteins for which two conformations are available[17]. DynDom analysis (Supplementary Table 1) revealed a small moving domain (Fig. 1b) that rotates by 7.1 degrees about the hinge axis from the larger fixed domain, and identified 25 hinge-bending residues. We selected two of these residues for design, Val35 and Lys37 (numbering based on Uniprot sequence P00509), because they are found on the flexible loop

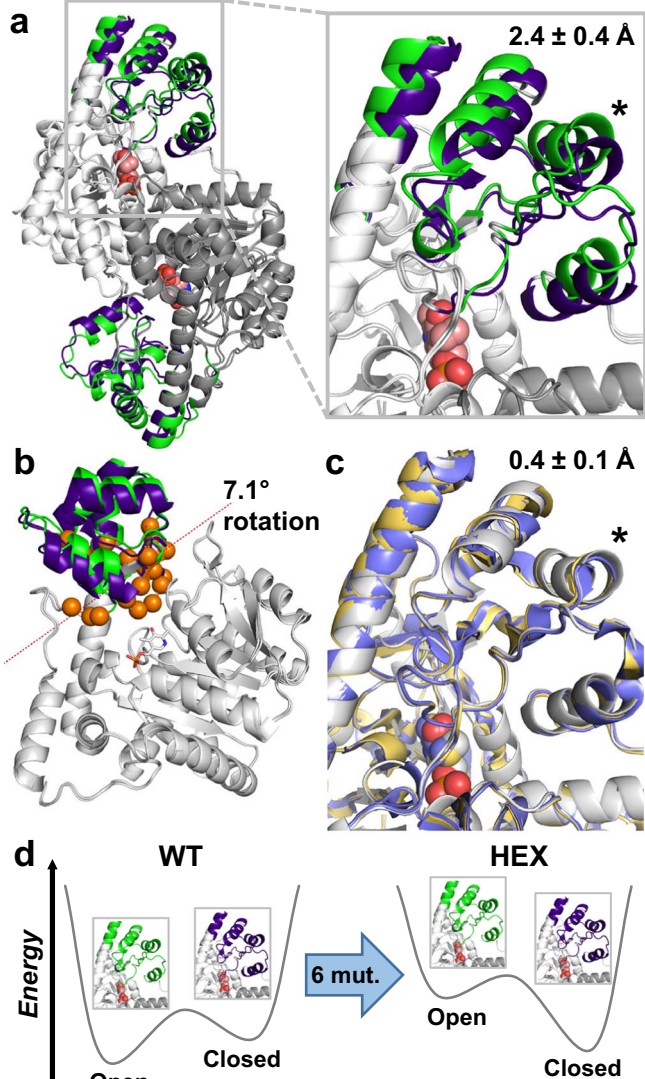

**Fig. 1 | AAT conformational landscape. a** *E. coli* AAT is a 90 kDa homodimer that undergoes a conformational change from an open (green, PDB ID: 1ARS) to closed (dark blue, PDB ID: 1ART) state upon substrate binding. This conformational transition involves rotation of a small moving domain (colored) relative to a fixed domain (white and gray for chains A and B, respectively), which causes a 2.4 ± 0.4 Å displacement (mean $C_\alpha$ distance ± s.d.) of the helix formed by residues K355−F365 (indicated by an asterisk). The PLP cofactor bound at the active site is shown as spheres (salmon). **b** Hinge movement analysis of chain A reveals a 7.1-degree rotation of the moving domain relative to the fixed domain along an axis between two planes (dotted line). Hinge-bending residues and PLP are shown as orange spheres and white sticks, respectively. **c** Superposition of HEX structures in the absence (yellow, PDB ID: 1AHE) and presence (blue, PDB ID: 1AHY) of bound inhibitor with that of the WT closed state (white, PDB ID: 1ART) show that this mutant is closed in both cases. $C_\alpha$ displacement (mean ± s.d.) of residues K355−F365 (asterisk) is indicated. **d** These results demonstrate that the six mutations (mut.) of HEX remodel its conformational landscape to favor the closed conformation.

**Table 1 | Apparent kinetic parameters of *E. coli* AAT and its mutants for transamination of various amino-acid donors with α-ketoglutarate as acceptor**

| Enzyme[a] | Mutations[b] | | | | L-Aspartate[c] | | | L-Phenylalanine[c] | | | Selectivity[d] |
|---|---|---|---|---|---|---|---|---|---|---|---|
| | V35 | K37 | T43 | N64 | $K_M$ (mM) | $k_{cat}$ (s$^{-1}$) | $k_{cat}/K_M$ (M$^{-1}$ s$^{-1}$) | $K_M$ (mM) | $k_{cat}$ (s$^{-1}$) | $k_{cat}/K_M$ (M$^{-1}$ s$^{-1}$) | |
| **Controls** | | | | | | | | | | | |
| WT | – | – | – | – | 0.21 ± 0.03 | 7.8 ± 0.3 | 37,000 ± 5000 | N.D.[f] | N.D.[f] | 400 ± 100 | 0.01 |
| HEX[e] | L | Y | I | L | 0.059 ± 0.007 | 1.32 ± 0.03 | 22,000 ± 3000 | 0.27 ± 0.03 | 9.0 ± 0.2 | 33,000 ± 4000 | 1.5 |
| **Closed Library** | | | | | | | | | | | |
| IYIT | I | Y | I | T | 0.014 ± 0.002 | 0.155 ± 0.005 | 11,000 ± 2000 | 0.34 ± 0.03 | 10.1 ± 0.2 | 30,000 ± 3000 | 2.7 |
| VFIT | – | F | I | T | 0.027 ± 0.003 | 0.90 ± 0.01 | 33,000 ± 4000 | 0.90 ± 0.06 | 22.8 ± 0.4 | 25,000 ± 2000 | 0.8 |
| VFIY | – | F | I | Y | 0.12 ± 0.01 | 0.263 ± 0.005 | 2200 ± 200 | 0.42 ± 0.04 | 17.3 ± 0.4 | 41,000 ± 4000 | 19 |
| VYIT | – | Y | I | T | 0.08 ± 0.01 | 0.37 ± 0.01 | 4600 ± 600 | 1.11 ± 0.07 | 41.8 ± 0.6 | 38,000 ± 2000 | 8 |
| VYIY | – | Y | I | Y | 0.09 ± 0.02 | 0.244 ± 0.006 | 2700 ± 600 | 0.58 ± 0.02 | 20.9 ± 0.2 | 36,000 ± 1000 | 13 |
| **Open$_{Low}$ Library** | | | | | | | | | | | |
| IFCA | I | F | C | A | 0.031 ± 0.004 | 2.63 ± 0.06 | 80,000 ± 10,000 | 2.3 ± 0.2 | 47 ± 1 | 20,000 ± 2000 | 0.25 |
| MFCA | M | F | C | A | 0.018 ± 0.003 | 1.21 ± 0.02 | 70,000 ± 10,000 | 1.08 ± 0.08 | 35.9 ± 0.6 | 33,000 ± 3000 | 0.47 |
| VFCA | – | F | C | A | 0.050 ± 0.006 | 4.4 ± 0.1 | 90,000 ± 10,000 | 4.2 ± 0.2 | 47 ± 1 | 11,200 ± 600 | 0.12 |
| VFCS | – | F | C | S | 0.068 ± 0.007 | 6.0 ± 0.1 | 88,000 ± 9000 | 12 ± 1 | 87 ± 3 | 7200 ± 700 | 0.08 |
| **Open$_{High}$ Library** | | | | | | | | | | | |
| AIFS | A | I | F | S | 0.32 ± 0.03 | 2.42 ± 0.04 | 7600 ± 700 | 2.6 ± 0.2 | 0.54 ± 0.01 | 210 ± 20 | 0.03 |
| CIFC | C | I | F | C | 0.13 ± 0.01 | 2.62 ± 0.04 | 20,000 ± 2000 | 5.9 ± 0.4 | 3.89 ± 0.08 | 660 ± 50 | 0.03 |
| CIFS | C | I | F | S | 0.28 ± 0.02 | 0.65 ± 0.01 | 2300 ± 200 | 5.4 ± 0.4 | 0.257 ± 0.006 | 48 ± 4 | 0.02 |
| SIFH | S | I | F | H | 0.33 ± 0.03 | 1.11 ± 0.02 | 3400 ± 300 | 3.7 ± 0.3 | 0.245 ± 0.008 | 66 ± 6 | 0.02 |
| SIFS | S | I | F | S | 0.35 ± 0.04 | 1.04 ± 0.02 | 3000 ± 300 | 3.9 ± 0.2 | 0.294 ± 0.006 | 75 ± 4 | 0.03 |

[a]Mutants are named on the basis of the amino-acid identity at the four designed positions. For example, the VFIT mutant from the Closed Library contains Val, Phe, Ile, and Thr residues at positions 35, 37, 43, and 64, respectively. WT and HEX refer to wild-type AAT and the previously published hexamutant[13], respectively.

[b]Mutations are numbered based on Uniprot sequence POO509. V35, K37, T43, and N64 correspond to V39, K41, T47, and N69 in the previously published crystal structures of wild-type AAT (PDB ID: 1ARS and 1ART).

[c]All experiments were performed in triplicate using a single enzyme batch. Errors of regression fitting, which represent the absolute measure of the typical distance that each data point falls from the regression line, are provided. Concentrations of α-ketoglutarate used to determine apparent kinetic parameters of donor substrates are reported on Supplementary Table 3.

[d]Selectivity is defined as ($k_{cat}/K_M$ L-phenylalanine)/($k_{cat}/K_M$ L-aspartate).

[e]This variant also contains the T104S and N285S mutations located near the cofactor phosphate moiety (T109S and N297S according to the numbering from HEX crystal structures [PDB ID: 1AHE and 1AHY]).

[f]Individual parameters $K_M$ and $k_{cat}$ could not be determined accurately because saturation was not possible at the maximum substrate concentration tested (40 mM, Supplementary Fig. 2), which is the substrate's solubility limit. Catalytic efficiency ($k_{cat}/K_M$) was therefore calculated from the slope of the linear portion ([S] « $K_M$) of the Michaelis-Menten model ($v_0 = (k_{cat}/K_M)[E_0][S]$).

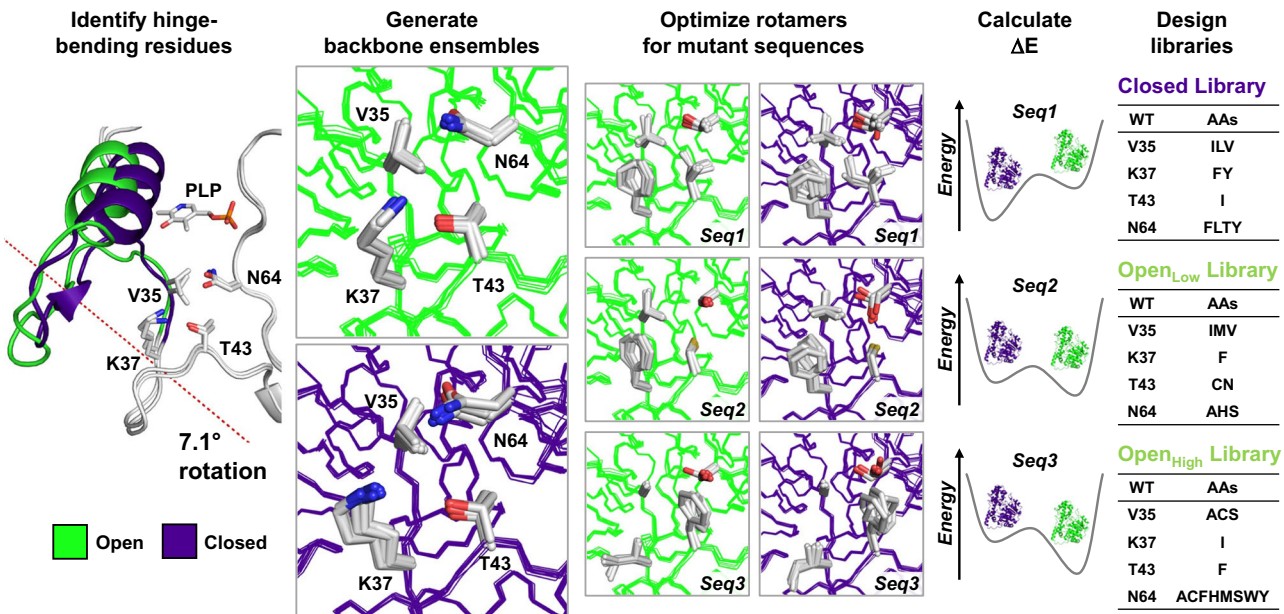

**Fig. 2 | Computational remodeling of AAT conformational landscape by multistate design.** To remodel the AAT conformational landscape, we followed a 5-step process: (1) identification of hinge-bending residues involved in transition between open (green) and closed (dark blue) conformational states; (2) generation of structural ensembles approximating backbone flexibility to model open and closed states; (3) optimization of rotamers for mutant sequences on both open- and closed-state ensembles; (4) calculation of energy differences between conformational states ($\Delta E = E_{closed} - E_{open}$) to predict equilibrium of each mutant, and (5) combinatorial library design using $\Delta E$ values to generate Closed, Open$_{Low}$, and Open$_{High}$ libraries for experimental testing. Designed residues V35, K37, T43, and N64 correspond to V39, K41, T47, and N69 in the previously published crystal structures of wild-type AAT (PDB ID: 1ARS and 1ART).

**Table 2 | Conformational equilibrium of AAT variants**

| Enzyme | Computational[a] | | | Experimental[b] | | | | |
|---|---|---|---|---|---|---|---|---|
| | $E_{closed}$ (kcal mol⁻¹) | $E_{open}$ (kcal mol⁻¹) | $\Delta E$ (kcal mol⁻¹) | $\Delta H$ (kcal mol⁻¹) | $\Delta S$ (kcal mol⁻¹ K⁻¹) | $\Delta C_p$ (kcal mol⁻¹ K⁻¹) | $\Delta G$, 278 K (kcal mol⁻¹) | $\Delta G$, 303 K (kcal mol⁻¹) |
| WT | −358.0 | −372.2 | 14.2 | N.D. | N.D. | N.D. | N.D. | N.D. |
| HEX | −350.2 | −273.4 | −76.8 | −9.9 | −0.032 | 0.972 | −1.72 | −0.31 |
| VFIT | −366.1 | −320.6 | −45.5 | −10.7 | −0.038 | 0.667 | −0.59 | 0.78 |
| VFIY | −357.8 | −300.0 | −57.8 | −19.6 | −0.070 | 0.369 | −0.38 | 1.59 |
| VFCS | −385.6 | −389.7 | 4.2 | N.D. | N.D. | N.D. | N.D. | N.D. |
| AIFS | −237.2 | −364.4 | 127.2 | N.D. | N.D. | N.D. | N.D. | N.D. |

[a]Boltzmann-weighted average potential energies ($T = 300$ K) for the open and closed state ensembles were computed using the Phoenix energy function for protein design (Methods section). Phoenix energies correspond to the sum of pairwise nonbonded interaction energies between rotamers and between rotamers and template in the folded state, without consideration of bonded interactions, entropy, or the energy of the unfolded state. The energy difference reported corresponds to $E_{closed} - E_{open}$.
[b]Differences in enthalpy ($\Delta H$), entropy ($\Delta S$), heat capacity ($\Delta C_p$), and Gibbs free energy ($\Delta G$) are given in the direction of enzyme closing (e.g., $G_{closed} - G_{open}$), and were calculated using a reference temperature of 298 K.
*N.D.* not determined.

connecting the moving and fixed domains (Fig. 2). We also selected for design residues Thr43 and Asn64, which are not part of the hinge, but whose side chains form tight packing interactions with those of Val35 and Lys37. Interestingly, these residues comprise four of the six positions that were mutated in HEX (Table 1), demonstrating that our analysis using only WT structures led to the identification of positions that contribute to controlling the open/closed conformational equilibrium in AAT.

Next, we generated backbone ensembles from the open- and closed-state crystal structures to approximate the intrinsic flexibility of these two conformational states using the PertMin algorithm[18], which we previously showed to result in improved accuracy of protein stability predictions when used as templates in multistate design[19]. Using the protein design software Phoenix[20,21], we optimized rotamers for all combinations of proteinogenic amino acids with the exception of proline at the four designed positions on each backbone ensemble, yielding Boltzmann-weighted average energies for 130,321 (19⁴) AAT

sequences that reflect their predicted stability on each conformational state. To identify mutant sequences that preferentially stabilized the closed conformation, we computed the energy difference between closed and open state ensembles ($\Delta E = E_{closed} - E_{open}$) for each sequence using the Phoenix potential energy function developed for protein design. Phoenix energies correspond to the sum of pairwise nonbonded interaction energies between rotamers and between rotamers and template in the folded state, without consideration of bonded interactions, entropy or the energy of the unfolded state. As a final step, we used these $\Delta E$ values as input to the CLEARSS library design algorithm[20] to generate a 24-member combinatorial library of AAT mutants predicted to favor the closed state (Closed library, Supplementary Table 2) with a range of values (−86.0 to −9.5 kcal mol⁻¹) encompassing that of HEX (−76.8 kcal mol⁻¹, Table 2). As controls, we also generated two libraries of sequences predicted to favor the open conformation (Supplementary Table 2): the Open$_{Low}$ library, which contains 18 sequences predicted to stabilize the open state with $\Delta E$

values (0.8–16.4 kcal mol⁻¹) comparable to that of the WT (14.2 kcal mol⁻¹, Table 2), and the Open_High library, which contains 24 sequences predicted to more strongly favor the open state due to substantial destabilization of the closed state by >120 kcal mol⁻¹. While we postulated that the Open_Low library would yield mutants with wild-type-like conformational landscapes and therefore similar catalytic efficiency and substrate selectivity, we hypothesized that Open_High library mutants would be less efficient than WT with both native and non-native substrates due to their strong destabilization of the closed conformation, which is the active form of the enzyme[15]. Thus, experimental characterization of these three mutant libraries, which comprise non-overlapping sequences (Fig. 2), allowed us to assess the ability of the ΔE metric to predict sequences with conformational landscapes favoring the open or closed states.

## Kinetic analysis of designs

We screened the three mutant libraries for transamination activity with the non-native substrate L-phenylalanine ("Methods" section) and selected the most active mutants from each library for kinetic analysis. All selected mutants catalyzed transamination of L-phenylalanine or L-aspartate with α-ketoglutarate, and displayed substrate inhibition with this acceptor substrate, as is the case for WT (Table 1, Supplementary Tables 3–4, and Supplementary Figs. 2–5). All Closed library mutants displayed catalytic efficiencies towards L-phenylalanine that were improved by approximately two orders of magnitude relative to WT (Table 1), comparable to HEX, and all were similarly or more active with this non-native substrate than with L-aspartate, in stark contrast with WT AAT, which prefers the native substrate by a factor of 100 (Supplementary Fig. 6). Furthermore, all Closed library mutants were less catalytically efficient with L-aspartate than the WT despite having lower $K_M$ values for this substrate, similar to HEX. These results are consistent with previous studies that showed that release of the oxaloacetate product resulting from L-aspartate transamination is dependent on a conformational change from the closed to the open state that is partially rate determining[22]. Meanwhile, Open_Low and Open_High mutants favored the native over the non-native substrate by up to 50-fold, similar to WT. Open_Low mutants had lower $K_M$ values and were more catalytically efficient than WT with both L-phenylalanine and L-aspartate, which could be due to the fact that these mutants have ΔE values similar to the WT but stabilize both the open and closed conformations by >10 kcal mol⁻¹ (Table 2 and Supplementary Table 2). This is not the case for Open_High mutants, which have similar $K_M$ values but are less catalytically efficient with the native substrate than WT, in agreement with our hypothesis that strong destabilization of the catalytically active closed conformation would result in less efficient catalysis. Overall, these kinetic results support the hypothesis that the change in substrate selectivity is linked to the conformational equilibrium previously suggested by the data from the HEX mutant[13,14].

## Structural analysis of designs

To provide structural information on the conformations adopted by AAT mutants, we turned to room-temperature X-ray crystallography, which provides insight into enzyme conformational ensembles under conditions that are relevant to catalysis[23] and free of potential distortions or conformational bias introduced by sample cryocooling[24]. We crystallized WT, HEX, and select variants from the Closed (VFIT and VFIY), Open_Low (VFCS), and Open_High (AIFS) libraries. All six enzymes yielded crystals under similar conditions (Supplementary Table 5), which could only be obtained in the presence of maleate (Supplementary Fig. 1d), an inhibitor that stabilizes the closed conformation when bound by both WT[25] and HEX[14]. To obtain structures in the absence of maleate, we applied a rigorous crystal soaking method to serially dilute and extract the inhibitor from the crystallized enzymes (Methods). We collected X-ray diffraction data at room temperature (278 K) for all variants with the exception of AIFS, which could only be

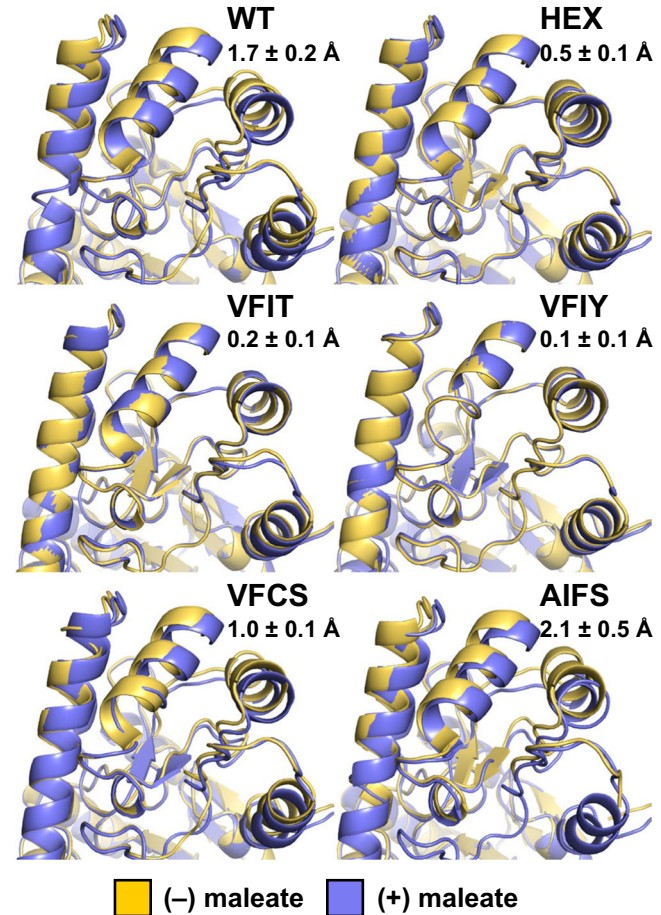

**Fig. 3 | Crystal structures.** Overlay of AAT structures (Chain A) in the presence and absence of maleate show transition from open to closed states upon inhibitor binding for WT, VFCS, and AIFS, but not for HEX, VFIT, and VFIY, which already adopt the closed conformation in the absence of bound inhibitor. Average displacements of helix formed by residues K355–F365 upon maleate binding are reported as the average pairwise distance of corresponding C_α atoms for the 11 residues comprising this helix (mean ± s.d.).

measured at cryogenic temperature (100 K) because we could only obtain small crystals that were not robust to radiation damage at non-cryogenic temperatures. We applied statistical criteria (Supplementary Table 6) to assign high-resolution cut-offs of 1.37–2.31 Å for our data sets, and all structures were subsequently determined by molecular replacement in space group P 6₃ with an enzyme homodimer in the asymmetric unit. Given the known impact of crystal packing and crystallization conditions on domain movement amplitude in AAT[25,26], we confirmed that crystals of all variants with or without soaking had similar unit cell dimensions and identical space group. In the inhibitor-bound state, all structures were closed as expected (Supplementary Fig. 7) due to electrostatic interactions between maleate and the side chains of Arg280 and Arg374 (Supplementary Fig. 8). Upon soaking crystals of the WT enzyme to remove the bound inhibitor, we observed that one subunit (chain A) within the enzyme homodimer was in the open conformation (Fig. 3), confirming that the soaking procedure was able to remove the bound maleate, and that the crystal lattice could accommodate the domain rotation required for opening and closing of this subunit. Chain B however remained closed upon maleate removal, likely due to crystal packing interactions hindering domain rotation of this subunit. These results are consistent with a previous study that showed that only one subunit of a homologous AAT closed upon soaking of the crystal in a substrate solution[26]. In the maleate-free structures, a sulfate ion and one or more ordered water molecules

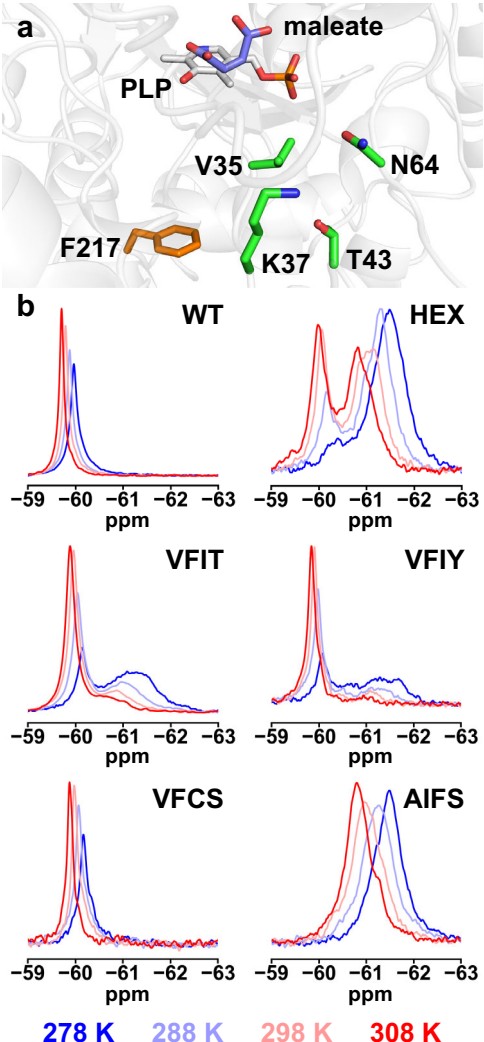

**Fig. 4 | Conformational landscape analysis by NMR. a** To evaluate conformational equilibrium of AAT variants, we introduced the fluorinated amino acid 4-trifluoromethyl-L-phenylalanine at position F217 (orange), which is located closer to hinge-bending and designed residues (green) than to the bound maleate inhibitor (blue). Crystal structure shown is that of wild-type (WT) AAT at 278 K (PDB ID: 8E9K). **b** $^{19}$F NMR spectra of AAT variants in the absence of ligand show dynamic equilibrium between 278 K and 308 K for HEX, VFIT, VFIY, and AIFS, confirming that these proteins are undergoing exchange. This is not the case for WT and Open$_{Low}$ library mutant VFCS, who both adopt predominantly the open conformation within this temperature range. For HEX and Closed library mutants VFIT and VFIY, the open conformation is enriched as temperature increases. For AIFS, spectra suggest that this Open$_{High}$ library mutant samples conformations distinct from those sampled by the other variants.

result suggests that the four mutations of AIFS, three of which are found within the Leu31–Thr43 loop, contribute to destabilize the open conformation, consistent with the calculated $E_{open}$ value of this variant (Table 2) being >7 kcal mol$^{-1}$ higher than that of the other variants that favor the open state (WT and VFCS). Interestingly, the amplitude of the open/closed conformational transition that occurs in open variants upon maleate binding (Fig. 3) correlated with their computed $\Delta E$ values (VFCS < WT < AIFS), suggesting that this metric can be used to fine-tune this enzyme conformational landscape. Furthermore, Dyn-Dom analyses of all variants confirmed that only WT, VFCS, and AIFS undergo hinge motion upon maleate binding (Supplementary Table 7), which rotates the moving domain relative to the fixed domain by 4.6, 2.6, and 5.9 degrees, respectively.

### NMR analysis of conformational landscapes

Having demonstrated crystallographically that our designed variants adopted the target conformation in the absence of ligand, we turned to NMR spectroscopy to gain insights into their conformational equilibria in solution, and compared results against those obtained for WT and HEX. We first measured $^{1}$H-$^{15}$N HSQC spectra for WT in the presence and absence of L-aspartate (Supplementary Fig. 11a). As expected for a protein of this size, the spectrum consisted of a large number of peaks that were broad with a high degree of overlap. It was nonetheless possible to assign the unique chemical shifts of peaks from the indole NH group for three native Trp residues by comparison of these spectra with those acquired with single Trp mutants (Supplementary Fig. 11b,c). This allowed assignment of a peak that was significantly broadened in spectra of both HEX and Closed library mutant VFIY to the indole NH from Trp307 (Supplementary Fig. 11d), whose side chain is closest to the designed hinge residues. Moreover, the Trp307 indole peak was no longer detectable when the L-aspartate substrate was present for both WT and mutant enzymes, most likely being broadened beyond detection. Since all conditions that favor the closed state (i.e., HEX and Closed library mutations and/or L-aspartate binding) broaden the Trp307 indole resonance, this exchange appears to be associated with the closed state, potentially due to conformational dynamics around the hinge.

In order to characterize the thermodynamics of the exchange processes around the hinge region, we labeled AAT variants at a single site with $^{19}$F using site-specific incorporation of the noncanonical amino acid 4-trifluoromethyl-L-phenylalanine[27]. Phe217 was chosen as the incorporation site since it is proximal to hinge residues but is not in direct contact with the substrate (Fig. 4a). The $^{19}$F spectrum of WT at 278 K showed a single peak centered at approximately –60 ppm that shifts downfield as temperature is increased (Fig. 4b), similar to what is observed for free 4-trifluoromethyl-L-phenylalanine in solution (Supplementary Fig. 12). By contrast, the HEX spectrum at 278 K showed a large broad peak centered at around –61.6 ppm, with another peak of substantially lower intensity also appearing at a similar shift to that observed in the WT spectrum (–60.4 ppm). The relative intensity of these 2 peaks changed as the temperature was increased, with the low-intensity peak increasing as the major peak decreased. This is characteristic of two-state exchange, with the equilibrium between the two states being shifted by the temperature change. Given that the crystal structure of HEX in its ligand-free form showed a closed conformation (Fig. 3), it is likely that the major peak reflects a local chemical environment created by the closed state, with a small population in an open state similar to that seen in the WT spectrum.

Using peak deconvolution and integration, it was possible to calculate relative populations for each species for variants undergoing the observed two-state exchange, along with the free energy difference between states (Supplementary Fig. 13 and Supplementary Table 8). We calculated $\Delta G$ at 278 K for HEX to be –1.72 kcal mol$^{-1}$ (Table 2), a small difference that would be compatible with the interconversion between these two states to be part of the catalytic cycle[28].

occupy the inhibitor binding site of both subunits (Supplementary Fig. 9), as was previously observed in WT[25] and HEX[14] structures at cryogenic temperatures.

Superposition of bound and unbound structures for each variant confirmed that closed library mutants VFIT and VFIY remained in the closed conformation in the inhibitor-free form, similar to HEX, while Open$_{Low}$ variant VFCS and Open$_{High}$ variant AIFS adopted the open conformation, similar to WT (Fig. 3). AIFS is unique in that the electron density is especially weak in the regions corresponding to the helix formed by Pro12–Leu19 and the loop connecting moving and fixed domains (Leu31–Thr43), even at cryogenic temperatures, demonstrating that these structural segments are disordered when the enzyme adopts the open conformation (Supplementary Fig. 10). This

These peak volumes could also be calculated over the entire temperature range tested, giving rise to a van't Hoff relationship with a small degree of curvature (Supplementary Fig. 14). Deviations from linearity can occur when there are differences in heat capacity between the two states, as would be expected for a process involving a change in conformational states over the temperature range tested[29]. By contrast, the presence of a single peak in WT spectra suggests that the closed state is not significantly populated under these conditions, as expected from its open-state crystal structures obtained under both cryogenic[12,25] and room-temperature conditions.

We next analyzed two mutants from the Closed library (VFIT and VFIY). [19]F NMR spectra at 278 K for both mutants showed a narrow peak centered at approximately –60.2 ppm that is similar to that of WT (Fig. 4). Spectra of these variants also showed another broad peak centered at approximately –61.2 ppm that is similar to the HEX peak characteristic of the closed conformation. The relative intensity of the two peaks showed similar temperature dependence, with an increase in the relative intensity of the WT-like peak as temperature was increased to 308 K. Peak deconvolution and integration (Supplementary Figs. 13–14 and Supplementary Table 8) were used to evaluate the population of the two states, and confirmed that at low temperatures both VFIT and VFIY favor the state resembling that adopted by the closed HEX mutant ($T < 288$ K or 283 K, respectively). However, unlike HEX, at higher temperatures the alternate conformation with the WT-like peak becomes the favored state (Table 2). Interestingly, $\Delta E$ values for these mutants were smaller in magnitude than that calculated for HEX, supporting the predictive nature of the calculated energy differences between open and closed states for these sequences.

To determine if these differences in the temperature dependence of exchange could be observed in the crystal state, we also solved the maleate-free structures of WT, HEX, and VFIT at 303 K (Supplementary Table 9 and Supplementary Fig. 9), and calculated isomorphous difference density maps by subtracting electron density at 278 K (Supplementary Fig. 15). Comparing VFIT data obtained at 278 K and 303 K results in substantial difference density throughout chain A, the chain that opens when WT crystals are soaked to remove maleate. By contrast, similar comparisons for WT and HEX showed relatively little difference density. This analysis confirms that VFIT undergoes larger local conformational changes than either HEX or WT when the temperature of the crystal is increased to 303 K, in agreement with our van't Hoff analysis of NMR data (Table 2, Supplementary Table 8). The agreement between temperature-dependent X-ray crystallography and NMR data provides strong evidence that the conformational exchange detected in the NMR experiments reflects a dynamic equilibrium between closed and open conformations, with mutations that favor the closed conformation also increasing selectivity toward L-phenylalanine.

We next analyzed two mutants designed to favor the open conformation in their ligand-free forms. Open_Low mutant VFCS showed [19]F NMR spectra that were very similar to those of WT within the tested temperature range (Fig. 4b and Supplementary Fig. 13), consistent with its preference for the open conformation (Fig. 3 and Table 2). However, Open_High mutant AIFS gave rise to spectra that were distinct from those of all other variants (Fig. 4b), but could be deconvoluted to two exchanging peaks (Supplementary Fig. 13), to allow estimation of populations (Supplementary Table 8 and Supplementary Fig. 14). We postulate that those peaks correspond to alternate open conformations distinct from the one sampled by the other variants. This hypothesis is supported by our observations that ligand-free AIFS is open at low temperature (Fig. 3) but contains disordered segments around the Pro12–Leu19 helix and the loop containing three of the four designed positions (Supplementary Fig. 10), which are located close to the Phe217 position where the [19]F label was introduced. The observed heterogeneity could therefore correspond to a mixture of these alternate open conformations. Additional support that the exchange

detected in AIFS differed from that of HEX and closed mutants was provided by the van't Hoff analysis, which showed no curvature for AIFS unlike for the closed mutants (Supplementary Fig. 14), suggesting that the Open_High variant does not undergo the open/closed conformational transition within this temperature range.

## Discussion

Here, we successfully remodeled the conformational landscape of an enzyme via targeted alterations to the equilibrium between two distinct conformational states related by a hinge-bending motion. The resulting equilibrium shift promoted activity toward a non-native substrate, leading to a selectivity switch of up to 1900-fold. Our approach should be applicable to the redesign of other enzymes where alterations to known conformational equilibria, including those mediated by hinge-bending motions, are expected or hypothesized to increase catalytic efficiency[7,30] or alter substrate selectivity[31]. As many enzymes undergo hinge-mediated domain motions during their catalytic cycles[32], the multistate design approach presented here should be straightforward to implement for such enzymes. Given the ability of our multistate design procedure to distinguish between closed and open states whose free energy difference is on the order of a single hydrogen bond, our approach could, in principle, also be applied to preferentially stabilize catalytically competent substates involving more subtle structural changes, such as backbone carbonyl flips[33] or side-chain rotations[34]. In such cases, it will be important to have structural information available for the target states. This methodology could therefore help to tailor catalytic efficiency or substrate selectivity by mimicking, in silico, the processes of evolution that harness altered conformational equilibria to tune function[7,35,36].

The predictive capacity of our multistate design framework could only be achieved by evaluating the energy of sequences on multiple conformational states. For example, the VFCS variant that prefers the open state is predicted to be more stable on the closed state than Closed library mutants VFIT and VFIY (Table 2), and more stable on the Open state than AIFS even though it is less open than this variant (Fig. 3). Furthermore, there were no obvious trends in designed mutations that could explain their effect on the conformational landscape, as none of these introduced bulkier or smaller amino acids at all or specific residue positions to cause or alleviate steric clashes in the open or closed conformations so as to shift the equilibrium towards one of these states, which has been the approach others have used to shift conformational equilibria[37]. Thus, subtle effects of mutation combinations on the relative stability of each conformational state were likely responsible for the observed preference of mutants for the open or closed conformations.

Our results demonstrate the utility of multistate design, with $\Delta E$ values calculated from ensemble energies of open and closed states, for the targeted alteration of subtle conformational equilibria, an approach that represents a useful alternative to heuristic methods that others have used to tune the relative stability of protein conformational states[38]. Extending this concept, we envision that de novo design of artificial enzymes with native-like catalytic efficiency and selectivity for complex multistep chemical transformations will require a holistic approach where every conformational state and/or substate required to stabilize reaction intermediates and transition states are explicitly modeled, and their relative energies optimized. The multistate design method for conformational landscape remodeling presented here opens the door to alter this common type of enzyme conformational equilibrium to facilitate the creation of designer biocatalysts with tailored functionality.

## Methods

### Structure preparation and ensemble generation

Crystal structures of wild-type *Escherichia coli* AAT in its internal aldimine form (PDB ID: 1ARS[12]) or complexed with *N*-phosphopyridoxyl-L-glutamic acid (PDB ID: 1X28[15]) were used to model the open

and closed states, respectively. To eliminate biases that could arise during ensemble generation from the presence of substrate bound in the active site, we deleted coordinates for the *N*-phosphopyridoxyl-L-glutamic acid and catalytic K246 residue in the 1X28 structure and replaced them with coordinates for K246 and the pyridoxal 5′-phosphate (PLP) cofactor extracted from the 1ARS structure. These structures were then prepared for ensemble generation using the Molecular Operating Environment (MOE) software[39]. Hydrogens were added with the Protonate3D utility and manually adjusted to ensure that the protonation states of PLP and K246 were consistent with the aminotransferase catalytic mechanism[40]. The resulting structures were then solvated in a rectangular box of water with counter ions (Na$^+$ and Cl$^-$) under periodic boundary conditions with a box cut-off of 6 Å, and energy minimized by conjugate gradient energy minimization to a root mean square gradient <7 kcal mol$^{-1}$ Å$^{-1}$ using the AMBER99 force field[41] with a combined explicit solvent and implicit reaction field solvent model set up using the MOE software package. These structures were used as input templates to generate backbone ensembles with the PertMin algorithm[18,19]. Briefly, two 50-member PertMin ensembles were created by randomly perturbing the coordinates of all heavy atoms of the two prepared AAT structures by ±0.001 Å along each Cartesian coordinate axis, and energy minimizing them using a truncated Newton[42] minimization algorithm for 100 iterations. The "Open" and "Closed" ensembles thus obtained displayed diversities (i.e. average backbone root mean square deviations between pairs of ensemble members) of 0.29 ± 0.03 Å and 0.32 ± 0.03 Å, respectively, and backbone root mean square deviations from the starting structure of 0.51 ± 0.02 Å.

## Computational protein design

All calculations were performed using the Phoenix protein design software[21,43] with the fast and accurate side-chain topology and energy refinement (FASTER) algorithm[44] for sequence optimization. The 2002 backbone-dependent Dunbrack rotamer library[45] with expansions of ±1 standard deviation around χ$_1$ and χ$_2$ was used to provide side-chain conformations of AAT residues to be threaded onto each fixed backbone template. Sequences were scored using the Phoenix energy function, a five-term potential energy function consisting of a Lennard-Jones 12−6 van der Waals term from the Dreiding II force field[46] with atomic radii scaled by 0.9, a direction-dependent hydrogen bond term with a well depth of 8.0 kcal mol$^{-1}$ and an equilibrium donor-acceptor distance of 2.8 Å[47], an electrostatic energy term modeled using Coulomb's law with a distance-dependent dielectric of 10, an occlusion-based solvation penalty term[21], and a secondary structural propensity term[48]. Sidechain rotamers of residues 35, 37, 43, and 64 were optimized on each backbone template using all proteinogenic amino acids with the exception of proline. Sidechain rotamers of residues within 5 Å of the designed residues were also optimized but their identities were not changed. The searched sequence space thus consisted of 130,321 (19$^4$) sequences, resulting in >13 million individual sequence energies (130,321 sequences × 50 backbones × 2 ensembles).

Boltzmann weighted average potential energies at 300 K were computed for each sequence on each ensemble, yielding energy values for the Closed and Open states ($E_{closed}$ and $E_{open}$). Energy differences between conformational states ($\Delta E = E_{closed} - E_{open}$) were then computed for use in library design. To avoid favoring sequences displaying unfavorable $E_{closed}$ and/or $E_{open}$ values resulting in favorable $\Delta E$ (e.g., high, low, or high absolute $\Delta E$ values for Closed, Open$_{Low}$, or Open$_{High}$ libraries, respectively), which would be expected to be unstable, sequences whose $E_{closed}$ and/or $E_{open}$ value fell outside of the 75$^{th}$ percentile were discarded.

## Library design

Library design was performed with the CLEARSS algorithm[20] using as input the $\Delta E$ values obtained as described above. For a specific library

size configuration, which is the specific number of amino acids at each position in the protein (e.g., 4 amino acids at position 1, 3 amino acids at position 2, etc.), the highest probability set of amino acids at each position were included in the library. To identify the optimal library size configuration, all configurations that lead to a combinatorial library of target sizes (in this case, 20 ± 4 sequences) were scored by taking the sum of all partition functions of the chosen amino acid sets over all positions, and the highest scoring library was selected. $E_{closed}$, $E_{open}$, and $\Delta E$ values for mutants from each library are reported on Supplementary Table 2.

## Chemicals

All reagents used were of the highest available purity. Synthetic oligonucleotides were purchased from Eurofins MWG Operon. Restriction enzymes and DNA-modifying enzymes were purchased from New England Biolabs. Ni-NTA agarose resin was purchased from Bio-Rad Laboratories. All aqueous solutions were prepared using water purified with a Barnstead Nanopure Diamond system.

## Mutagenesis

The wild-type *E. coli* AAT gene (Uniprot ID: P00509) with an N-terminal His-tag cloned into plasmid pET-45b (Novagen) via the *Nco*I/*Pac*I restriction sites[49] was a generous gift from Michael D. Toney (University of California, Davis). Mutations were introduced into the AAT gene by overlap extension mutagenesis[50] using VentR DNA Polymerase. Briefly, external primers containing *Nde*I or *Bam*HI restriction sites were used in combination with sets of complementary pairs of oligonucleotides containing mutated codons (individual codons for HEX and single point mutants, codon mixtures for the Closed, Open$_{Low}$, and Open$_{High}$ libraries) in individual polymerase chain reactions (PCRs). The resulting overlapping fragments were gel-purified (Omega Biotek) and recombined by overlap extension PCR. The resulting amplicons were digested with *Nde*I/*Bam*HI, gel-purified, and ligated into the pET-11a expression vector (Novagen) with T4 DNA ligase. pBAD vectors (Invitrogen) harboring selected AAT genes flanked by *Nco*I/*Eco*RI restriction sites were prepared using a similar procedure. All constructs were verified by sequencing the entire open reading frame. Amino-acid sequences of all AAT variants are listed in Supplementary Table 10, and a multiple sequence alignment is shown on Supplementary Fig. 16.

## Preparation of clarified cell lysates

DNA libraries prepared as described above were transformed into chemically competent *E. coli* BL21-Gold (DE3) cells (Agilent). Colonies (180 per library) were picked into individual wells of V96 MicroWell polypropylene plates (Nunc) containing 300 μL of lysogeny broth (LB) supplemented with 100 μg mL$^{-1}$ ampicillin and 10% glycerol. The plates were covered with a sterile breathable rayon membrane (VWR) and incubated overnight at 37 °C with shaking. After incubation, these mother plates were used to inoculate sterile Nunc V96 MicroWell polypropylene plates ("daughter" plates) containing 300 μL per well of Overnight Express Instant TB medium (Novagen) supplemented with ampicillin. Daughter plates were sealed with breathable membranes and incubated overnight (37 °C, 250 rpm). After incubation, cells were harvested by centrifugation (3000×*g*, 30 min, 4 °C) and pellets were washed twice with phosphate-buffered saline (pH 7.4). Washed cell pellets were resuspended in lysis buffer (100 mM potassium phosphate buffer pH 8.0 containing 1× Bug Buster Protein Extraction Reagent [Novagen], 5 U mL$^{-1}$ Benzonase Nuclease [EMD], and 1 mg mL$^{-1}$ lysozyme). Clarified lysates were collected following centrifugation and stored at 4 °C until used in the screening assay.

## Library screening

All assays were performed in 200-μL reactions at 37 °C in 100 mM potassium phosphate buffer (pH 8.0). The standard reaction mixture

contained final concentrations of either 3 or 40 mM L-phenylalanine, 16 μM PLP, 0.2 mM α-ketoglutarate, 1 U of glutamate dehydrogenase (GDH) from bovine liver (Sigma), and 5 mM NAD$^+$. Plates containing the standard reaction mixture were incubated at 37 °C for 5 min prior to initiation of the reaction by addition of 10 μL of clarified cell lysates prepared as described above. Enzyme reactions were monitored by measuring absorbance of NADH at 340 nm every 12 sec for 30 or 60 min in individual wells of 96-well plates (Greiner Bio-One) using a SpectraMax 384 Plus plate reader (Molecular Devices). The four or five most active variants from each library were selected for further characterization.

## Protein expression and purification

Selected mutants were expressed and purified as described by Mironov et al. [51]. Briefly, *E. coli* BL21-Gold (DE3) cells harboring expression vectors containing aminotransferase genes were grown at 37 °C in 500 mL LB medium supplemented with 100 μg mL$^{-1}$ ampicillin until they reached an OD600 of 0.6. Isopropyl β-D-1-thiogalactopyranoside (1 mM) was added to the flasks to induce protein expression, followed by shaking overnight at 16 °C. Cells were harvested by centrifugation, resuspended in 10 mL lysis buffer (5 mM imidazole in 100 mM potassium phosphate buffer, pH 8.0), and lysed with an EmulsiFlex-B15 cell disruptor (Avestin). Proteins were purified by immobilized metal affinity chromatography using Ni–NTA agarose pre-equilibrated with lysis buffer in individual Econo-Pac gravity-flow columns (Bio-Rad). Columns were washed twice, first with 10 mM imidazole in 100 mM potassium phosphate buffer (pH 8.0), and then with the same buffer containing 20 mM imidazole. Bound proteins were eluted with 250 mM imidazole in 100 mM potassium phosphate buffer (pH 8.0) and exchanged into 100 mM sodium phosphate buffer (pH 8.0) using Econo-Pac 10DG desalting pre-packed gravity flow columns (Bio-Rad). For crystallography, proteins were further purified by gel filtration in 20 mM potassium phosphate buffer (pH 7.5) using an ENrich SEC 650 size-exclusion chromatography column (Bio-Rad). Purified samples were concentrated using Microsep Advance 10 K centrifugal devices (Pall) to a final concentration of 170 μM. Protein concentrations were quantified using a modified version of the Bradford assay, where the calibration curve is constructed as a plot of the ratio of the absorbance measurements at 590 and 450 nm versus concentration[52].

## Steady-state kinetics

To measure steady-state kinetics for the α-ketoglutarate acceptor substrate, assays were performed by varying the α-ketoglutarate concentration from 0.002 to 20 mM in the presence of 10–20 mM L-phenylalanine or L-aspartate (Supplementary Table 4), 5 mM NAD$^+$, 16 μM PLP, 1 U GDH, and approximately 10 mU of aminotransferase in 100 mM potassium phosphate buffer (pH 8, 37 °C). To measure steady-state kinetics for the L-aspartate and L-phenylalanine donor substrates, assays were performed by varying the amino-acid concentration from 0.002 to 40 mM in the presence of 0.1563–1.25 mM α-ketoglutarate (Supplementary Table 3), 5 mM NAD$^+$, 16 μM PLP, 1 U GDH, and approximately 10 mU of aminotransferase in 100 mM potassium phosphate buffer (pH 8, 37 °C). The pH of all reaction mixtures was adjusted to 8.0 prior to initiation of the reaction. Enzyme reactions were monitored by measuring absorbance of NADH at 340 nm ($\varepsilon = 6220$ M$^{-1}$ cm$^{-1}$) every 12 sec for 30 or 60 min in individual wells of 96-well plates (Greiner Bio-One) using a SpectraMax 384 Plus plate reader (Molecular Devices). Path lengths for each well were calculated ratiometrically using the difference in absorbance of potassium phosphate buffer at 900 and 998 nm. Linear phases of kinetic traces were used to measure initial reaction rates. Initial reaction rates at different substrate concentrations were fit to the Michaelis-Menten equation using Python ver.2.7.15 with the *scipy.optimize.curve fit* function (scipy ver.1.1.0). For mutant and substrate combinations resulting in substrate inhibition, fitting of the kinetic data was done

with a rate equation that takes into account this type of inhibition: $v_0 = (v_{max}[S])/(K_M + [S] + [S]^2/K_i)$.

## Preparation of $^{15}$N labeled proteins

Proteins for NMR spectroscopy were expressed using M9 minimal expression medium supplemented with 1 g L$^{-1}$ $^{15}$N-labeled ammonium chloride ($^{15}$NH$_4$Cl) for isotopic enrichment. Cultures were grown at 37 °C with shaking to an optical density at 600 nm of approximately 0.6, after which protein expression was initiated with 1 mM isopropyl β-D-1-thiogalactopyranoside. Following overnight incubation at 16 °C with shaking (275 rpm), cells were harvested by centrifugation and lysed with an EmulsiFlex-B15 cell disruptor (Avestin). Proteins were purified by immobilized metal affinity chromatography as described above, which was followed by gel filtration in 10 mM sodium phosphate buffer (pH 6.0) using an ENrich SEC 650 size exclusion chromatography column (Bio-Rad). Purified samples were concentrated using Amicon Ultracel-10K centrifugal filter units (EMD Millipore).

## Preparation of $^{19}$F site-specific labeled proteins

Site-specific incorporation of 4-trifluoromethyl-L-phenylalanine into selected AAT variants was prepared with a protocol adapted from Hammill et al. [27]. Briefly, chemically competent *E. coli* DH10B cells (Thermo Fisher) harboring the pDule-4-tfmF A65V S158A plasmid (Addgene plasmid #85484)[53], which encodes an orthogonal aminoacyl-tRNA synthetase and cognate amber suppressing tRNA for site-specific incorporation of 4-trifluoromethyl-L-phenylalanine, were transformed with pBad vectors (Invitrogen) containing mutated genes of AAT variants in which an amber stop codon was introduced at residue position 217 to allow direct incorporation of the fluorinated amino acid. Labeled AAT variants were expressed in 2 L of LB containing 100 μg mL$^{-1}$ ampicillin and 15 μg mL$^{-1}$ tetracycline at 37 °C. After cells were grown for 1 h, 468 mg of 4-trifluoromethyl-L-phenylalanine (SynQuest Laboratories) was added to the flask to give a final concentration of 1 mM. Once the cell culture reached an OD600 of 0.6, L-arabinose was added to a final concentration of 0.2% to induce protein expression. Following overnight incubation at 16 °C with shaking, cells were harvested by centrifugation, resuspended in 10 mL lysis buffer, and lysed with an EmulsiFlex-B15 cell disruptor (Avestin). Proteins were then extracted and purified by immobilized metal affinity chromatography, as described above. Elution fractions containing the aminotransferases were concentrated through centrifugation (Pall Microsep Advance Centrifugal Device 10 K), resuspended in 10 mM potassium phosphate buffer pH 6.0 and further purified through size exclusion chromatography, as described above. Elution fractions containing the purified aminotransferase were combined and concentrated through centrifugation.

## $^1$H-$^{15}$N heteronuclear single quantum coherence spectroscopy

$^{15}$N-labeled AAT samples for NMR (wild type, HEX, VFIY, and single point mutants Trp124Phe, Trp194Phe, Trp307Phe) consisted of 0.2–0.6 mM protein in 10 mM sodium phosphate buffer (pH 7.4), 10 μM EDTA, 0.02% sodium azide, and 10% D$_2$O. HSQC experiments were performed on a Bruker AVANCEIII HD 600 MHz spectrometer equipped with a triple resonance cryoprobe. 128 scans were accumulated to acquire each spectrum.

## $^{19}$F nuclear magnetic resonance spectroscopy

Protein samples used for $^{19}$F NMR analysis were diluted to concentrations of 75–280 μM in 500 μL of 10 mM potassium phosphate buffer (pH 6.0), 100 μM EDTA, 0.02% sodium azide, and 10% D$_2$O. All $^{19}$F NMR spectra were acquired with a Bruker Avance 500 MHz spectrometer with 10 sec acquisition delays. 512 scans were accumulated for one-dimensional $^{19}$F chemical shift analysis at 5, 10, 15, 20, 25, 30, and 35 °C. Data were processed with an exponential window function (20 Hz line-broadening) using TopSpin 3.6.1 (Bruker Biospin). The $^{19}$F NMR spectra

of fluorinated HEX, VFIT, and VFIY showed two resonances, which we interpret to correspond to two distinct conformations that do not rapidly exchange with each other on the $^{19}$F-NMR timescale. All spectra were deconvoluted into two Lorentzian curves using Python v3.7.9 with the LMFIT package for nonlinear least-squares minimization and curve-fitting[54]. The resulting two curves were integrated to measure the relative populations of each conformational state. The ratio of the two populations were used to calculate equilibrium constants at each temperature (Supplementary Table 8) and fit using the Solver function of Excel to the nonlinear van't Hoff equation[55] (Equation 1), where enthalpy and entropy are not assumed to be temperature-independent (i.e. $\Delta C_p \neq 0$) to extract thermodynamic parameters of equilibrium. When fitting to the linear van't Hoff equation, $\Delta C_P$ was simply set to zero.

$$\ln\left(K_{eq}\right) = -\frac{\Delta H^{\circ}_{ref}}{R}\left(\frac{1}{T}\right) + \frac{\Delta S^{\circ}_{ref}}{R} - \frac{\Delta C_P}{R}\left[\left(\frac{T-T_{ref}}{T}\right) + \ln\left(\frac{T_{ref}}{T}\right)\right]$$

Equation 1. Nonlinear van't Hoff equation. $T_{ref}$ is an arbitrary reference temperature (set to 298 K), $\Delta H^{\circ}_{ref}$ and $\Delta S^{\circ}_{ref}$ are $\Delta H^{\circ}(T)$ and $\Delta S^{\circ}(T)$ evaluated at $T_{ref}$, respectively.

## Crystallization

Purified AAT variants were prepared in 20 mM potassium phosphate buffer (pH 7.5) with 2 mM EDTA and 10 uM PLP to a final concentration indicated on Supplementary Table 5. Maleic acid was dissolved in water to make a 1 M stock solution, and then added to each protein solution yielding a final maleate concentration of 20 mM. For each enzyme variant, we carried out initial crystallization trials in 15-well hanging drop format using EasyXtal crystallization plates (NeXtal) and a crystallization screen that was designed to explore parameter space around the crystallization conditions reported by Islam et al.[15]. Crystallization drops were prepared by mixing 1 μL of protein solution with 1 μL of the mother liquor and sealing the drop inside a reservoir containing 500 μL of mother liquor. The mother liquor solutions contained ammonium sulfate as a precipitant and the specific growth conditions that yielded the crystals used for X-ray data collection are provided in Supplementary Table 5. For all six enzymes, a microseeding protocol was required to obtain high-quality crystals. Microseeds were prepared by crushing initial crystals in their mother liquor using a glass rod, and were subsequently streaked into the crystallization drops using a cat whisker. Because maleate was required for crystallization, ligand-free structures were obtained by soaking the crystal in new drops of 1 μL mother liquor containing 10 μM PLP but no maleate, allowing maleate in the crystal to diffuse out. Each crystal was treated in this way six subsequent times, 12 hours apart, to achieve removal of bound maleate (Supplementary Fig. 9).

## X-ray data collection and processing

Prior to X-ray data collection, crystals were mounted on polymer MicroMounts (MiTeGen) and sealed using a MicroRT tubing kit (MiTeGen). Single-crystal X-ray diffraction data was collected on beamline 8.3.1 at the Advanced Light Source. The beamline was equipped with a Pilatus3 S 6 M detector and was operated at a photon energy of 11111 eV. Crystals were maintained at either 100 K, 278 K, or 303 K throughout the course of data collection. Each data set was collected using a total X-ray dose of 50–100 kGy and covered a 180° wedge of reciprocal space. Multiple data sets were collected for each enzyme variant either from different crystals, or if their size permitted, from unique regions of single large crystals.

X-ray data were processed with the Xia2 program[56], which performed indexing and integration with DIALS[57], followed by scaling with DIALS.SCALE[58]. The resolution cut-off was taken where the $CC_{1/2}$ and $\langle I/\sigma I\rangle$ values for the intensities fell to approximately 0.5 and 1.0 respectively.

## Structure determination

We obtained initial phase information for calculation of electron density maps by molecular replacement using the program Phaser[59], as implemented in v1.17.1.3660 of the PHENIX suite[60], with the crystal structure of wild-type *Escherichia coli* AAT complexed with *N*-phosphopyridoxyl-L-glutamic acid (PDB ID: 1X28[15]) as search model. All AAT variants crystallized in the same crystal form, containing two chains of the molecule in the crystallographic asymmetric unit. Next, we rebuilt the initial model using the electron density maps calculated from molecular replacement. We then performed additional, iterative refinement of atomic positions, individual atomic displacement parameters (B-factors), and occupancies using a translation-libration-screw (TLS) model, a riding hydrogen model, and automatic weight optimization, until the model reached convergence. All model building was performed using Coot 0.8.9.2[61] and refinement steps were performed with phenix.refine (v1.17.1.3660) within the PHENIX suite[60,61]. Restraints for PLP in its internal aldimine form were generated using phenix.elbow[62], starting from coordinates available in the Protein Data Bank[63] (PDB ligand ID: PLP), and manually edited to ensure planarity of the pyridine ring and correct geometry of the imine bond between PLP and K246 (restraints available as Supplementary Data Files). Further information regarding model building and refinement, as well as PDB accession codes for the final models, are presented in Supplementary Table 6 and Supplementary Table 9.

## Reporting summary

Further information on research design is available in the Nature Portfolio Reporting Summary linked to this article.

## Data availability

Structure coordinates for all AAT variants have been deposited in the RCSB Protein Data Bank with the following accession codes: 8E9C, 8E9D, 8E9J, 8E9K, 8E9L, 8E9M, 8E9N, 8E9O, 8E9P, 8E9Q, 8E9R, 8E9S, 8E9T, 8E9U, and 8E9V. NMR data has been deposited in the BMRbig Biological Magnetic Resonance Data Bank with accession code bmrbig92. Other relevant data are available from the corresponding author upon request. Source data are provided with this paper.

## Code availability

Phoenix scripts are available from the corresponding author upon request. Requests for the Phoenix protein design software should be addressed to Stephen Mayo (Caltech).

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

## Acknowledgements

R.A.C. acknowledges an Early Researcher Award from the Ontario Ministry of Economic Development & Innovation (ER14-10-139), and grants from the Natural Sciences and Engineering Research Council of Canada (RGPIN-2016-04831) and the Canada Foundation for Innovation (26503). The authors thank Michael D. Toney for providing the wild-type *E. coli* AAT expression vector and James S. Fraser for access to crystallization and beamline facilities. Beamline 8.3.1 at the Advanced Light Source is operated by the University of California San Francisco with generous support from the National Institutes of Health (R01 GM124149 for technology development and P30 GM124169 for user support), and the Integrated Diffraction Analysis Technologies program of the US Department of Energy Office of Biological and Environmental Research. The Advanced Light Source (Berkeley, CA) is a national user facility operated by Lawrence Berkeley National Laboratory on behalf of the US Department of Energy under contract number DE-AC02-05CH11231, Office of Basic Energy Sciences. Use of the Stanford Synchrotron Radiation Lightsource, SLAC National Accelerator Laboratory, is supported by the U.S. Department of Energy, Office of Science, Office of Basic Energy Sciences under Contract No. DE-AC02-76SF00515. The SSRL Structural Molecular Biology Program is supported by the DOE Office of Biological and Environmental Research, and by the National Institutes of Health, National Institute of General Medical Sciences (including P41GM103393). The contents of this publication are solely the responsibility of the authors and do not necessarily represent the official views of NIGMS or NIH.

## Author contributions

A.D.S. and R.A.C. conceived the project. A.D.S. performed computational design experiments. A.D.S. and M.G.E. purified proteins. A.D.S., S.M.F., and S.T.K. performed mutagenesis and cloning experiments. A.D.S. performed enzyme kinetics experiments. A.D.S. and A.M.D. performed NMR experiments. A.D.S. and N.K.G. designed NMR experiments. J.M.R. and M.C.T. crystallized proteins and performed X-ray diffraction experiments. R.A.C. and J.M.R. performed refinements. M.C.T. designed X-ray crystallography experiments. R.A.C. and A.D.S. wrote the manuscript. N.K.G., A.M.D., and M.C.T. edited the manuscript.

## Competing interests

The authors declare no competing interests.
