## [Peer Review File · Nature Communications]

[Mentions of prior referee reports have been redacted.]

REVIEWER COMMENTS

Reviewer #1 (Remarks to the Author):

The article describes a careful analysis of protein dynamics in catalysis that is relevant for enzyme engineers as well as structural biologists and should be made available to the scientific community, which is why I encourage its publication. The authors have addressed all questions and concerns of the earlier reviews and adjusted the manuscript accordingly, so I support publication as is.

Reviewer #2 (Remarks to the Author):

I believe that Nat Commun is a more appropriate journal for this work than [another Nature journal]. That, combined with tempering of the general claims made, have addressed all of my concerns. This is a really elegant study and I think it will be of great interest to the structural biology and molecular biophysics communities.

Reviewer #3 (Remarks to the Author):

This sentence is the primary justification for the current work and is a bit contracted. A more complete discussion is in order:

73 Previously, a hexamutant (HEX) of AAT that is approximately two orders of magnitude more catalytically efficient than the wild type (WT) with the aromatic amino acid L-phenylalanine (Table 1), and similarly efficient for transamination of L-aspartate, was engineered by replacing six of the residues that are strictly conserved in AAT enzymes by those found at corresponding positions in the homologous E. coli tyrosine aminotransferase¹³.

The authors have optimized Asp-AT substrate selectivity based on the work of Onuffer and Kirsch. They describe the mutations made by Kirsch to be hinge mutations as AAT has a small domain that rotates in response to substrate binding. The mutations are actually active site residues which line the substrate channel. Kirsch was attempting to change AAT activity from aspartate to tyrosine. The Kirsch study was justified since the tyrosine aminotransferase structures were unknown. They based the site directed mutations on multiple sequence alignment in hopes of converting Asp-AT into Tyr-AT

The largest motion upon substrate binding is Leu 14 and Ile 33, two residues which occlude the substrate channel. The small domain motion (10 – 40 and 330 – C-term) is in response to this conformational change. The N-terminal residues are tethered in a domain swap.

The discussion about open and closed states should be clearer. The structures of Jager and Jansonius of E. coli AAT (PDB 1asl and 1asn) shows that chain A has hindered motion and chain B has restricted motion. This is supposedly caused by two factors, crystal contacts and high salt crystallization conditions (there is an SO₄ anion in the Michaelis site in both apo active sites). Superposition of B onto A shows that B is partially open (not fully closed). Interestingly, neither of these forms are fully open. The AAT crystal forms, pdb 1ajr and pdb 1ajs, show that unhindered motion of the small domain is significantly more. (Compare Fig 1 Rhee JBC 1997 vol 272, 17293. to Fig 3 in the Jager paper (ref 25)).

56 Using our approach, we enriched the less populated but catalytically active closed conformation in order to increase catalytic efficiency (k_{cat}/K_M) with the non-native substrate L-phenylalanine, leading to altered substrate selectivity.

If the point of this study was to improve substrate selectivity and the model chosen was to convert aspartate aminotransferase to tyrosine aminotransferase. Why choose malate as the substrate in co-crystals? Why not p-hydroxyphenylpyruvate?

Supplemental Figures 8, 9, 10, and 15 need the associated PDB IDs for clarity.

The Kirsch HEX study uses different residues numbers (V39L, K41Y, T47I, N69L, T109S, N297S).

Stating “numbering based on Uniprot sequence P00509” is fine but at least give the reader the comparison, V35L, K37Y, T43I, T104S, and N285S. The last two are dismissed without discussion? 825 d This variant also contains the T104S and N285S active-site mutations. These are all active site mutants.

The sequences in Supplement Table 10 should be a multiple-alignment with the N-termini indicated. Individual sequences are not very useful.

The PDB residue code for the LYS-PLP internal aldimine is LLP. This will help distinguish the structures in which PLP has become disconnected from K246. The malate substrate resides in the Michaelis site.

For simplicity, the structures should use the same crystallographic origin. VFIT apo, VFIY apo, HEX inhib, VFIT inhib, VFIY inhib, and VFCS inhib have different transform matrices. Not crucial but makes comparison easier.

Reviewers' Comments:

Reviewer #3 (Remarks to the Author):

This sentence is the primary justification for the current work and is a bit contracted. A more complete discussion is in order:

73 Previously, a hexamutant (HEX) of AAT that is approximately two orders of magnitude more catalytically efficient than the wild type (WT) with the aromatic amino acid L-phenylalanine (Table 1), and similarly efficient for transamination of L-aspartate, was engineered by replacing six of the residues that are strictly conserved in AAT enzymes by those found at corresponding positions in the homologous *E. coli* tyrosine aminotransferase¹³.

The authors have optimized Asp-AT substrate selectivity based on the work of Onuffer and Kirsch. They describe the mutations made by Kirsch to be hinge mutations as AAT has a small domain that rotates in response to substrate binding. The mutations are actually active site residues which line the substrate channel. Kirsch was attempting to change AAT activity from aspartate to tyrosine. The Kirsch study was justified since the tyrosine aminotransferase structures were unknown. They based the site directed mutations on multiple sequence alignment in hopes of converting Asp-AT into Tyr-AT

Response: We have expanded our justification of the current work as follows:

Page 4, Line 72: “Previously, AAT was redesigned to change its substrate specificity to allow transamination of aromatic amino acids [Onuffer Prot Sci 1995]. To do so, six of the 19 residues that are strictly conserved in AAT enzymes were replaced by those found at corresponding positions in the homologous *E. coli* tyrosine aminotransferase. These residues were selected because they line the substrate channel leading to the catalytic pocket or are located near the cofactor phosphate moiety. This process yielded a hexamutant (HEX) of AAT that is approximately two orders of magnitude more catalytically efficient than the wild type (WT) with the aromatic amino acid L-phenylalanine (Table 1), and similarly efficient for transamination of L-aspartate.

The largest motion upon substrate binding is Leu 14 and Ile 33, two residues which occlude the substrate channel. The small domain motion (10 – 40 and 330 – C-term) is in response to this conformational change. The N-terminal residues are tethered in a domain swap.

Response: We agree that Leu14 and Ile33 undergo large motion upon substrate binding in WT (but not in HEX or our Closed mutants). However, it is unclear whether substrate binding causes a conformational change of these residues' side chains, which in turn causes the small domain motion, or whether substrate binding causes the small domain motion, which in turn causes these residues to adopt a different side chain conformation. Our data alone does not allow us to delineate between these two possibilities, which is why we prefer not to discuss it.

The discussion about open and closed states should be clearer. The structures of Jager and Jansonius of *E. coli* AAT (PDB 1asl and 1asn) shows that chain A has hindered motion and

chain B has restricted motion. This is supposedly caused by two factors, crystal contacts and high salt crystallization conditions (there is an SO₄ anion in the Michaelis site in both apo active sites). Superposition of B onto A shows that B is partially open (not fully closed). Interestingly, neither of these forms are fully open. The AAT crystal forms, pdb 1ajr and pdb 1ajs, show that unhindered motion of the small domain is significantly more. (Compare Fig 1 Rhee JBC 1997 vol 272, 17293. to Fig 3 in the Jager paper (ref 25)).

Response: Thank you for bringing these structures to our attention. These structures (and associated articles) demonstrate that crystal packing and crystallization conditions have an impact on the amplitude of domain movement in AAT, justifying our efforts to solve our own structures of bound and unbound WT and HEX under similar crystallization conditions to (1) avoid crystal packing differences and thereby (2) evaluate opening/closing of our designed mutants in an unbiased way. All our enzymes crystallized under unit cells of similar size and identical space group. The observation that only one subunit of pig cytosolic AAT (1ajr/1ajs) closes when the unbound crystal is soaked with the methylaspartate substrate (Rhee JBC 1997) is in agreement with our results that show that only Chain A of E. coli AAT opens when the maleate inhibitor is removed via soaking. To clarify, we have modified the text as follows:

Page 9, line 186: “We applied statistical criteria (Supplementary Table 6) to assign high-resolution cut-offs of 1.37–2.31 Å for our data sets, and all structures were subsequently determined by molecular replacement in space group P 6₃ with an enzyme homodimer in the asymmetric unit. Given the known impact of crystal packing and crystallization conditions on domain movement amplitude in AAT [Jäger, Rhee], we confirmed that crystals of all variants with or without soaking had similar unit cell dimensions and identical space group.”

Page 9, line 194: “Upon soaking crystals of the WT enzyme to remove the bound inhibitor, we observed that one subunit (chain A) within the enzyme homodimer was in the open conformation (Figure 3), confirming that the soaking procedure was able to remove the bound maleate, and that the crystal lattice could accommodate the domain rotation required for opening and closing of this subunit. Chain B however remained closed upon maleate removal, likely due to crystal packing interactions hindering domain rotation of this subunit. These results are consistent with a previous study that showed that only one subunit of a homologous AAT closed upon soaking of the crystal in a substrate solution [Rhee JBC 1997].”

56 Using our approach, we enriched the less populated but catalytically active closed conformation in order to increase catalytic efficiency (kcat/KM) with the non-native substrate L-phenylalanine, leading to altered substrate selectivity.

If the point of this study was to improve substrate selectivity and the model chosen was to convert aspartate aminotransferase to tyrosine aminotransferase. Why choose malate as the substrate in co-crystals? Why not p-hydroxyphenylpyruvate?

Response: We used maleate (not malate) for crystallography to solve structures of the closed state for all variants because it has been shown to induce closure of both WT (PDB ID: 1ASM) and HEX (PDB ID: 1AHY). This made us confident that we could use maleate to induce closure of all our designed mutants, regardless of whether we designed them to favor the open or closed

state in the absence of ligand. We did not use p-hydroxyphenylpyruvate because we could not find any crystal structure of WT AAT bound to this ligand (or phenylpyruvate) in the PDB. To clarify, we modified this sentence:

“All six enzymes yielded crystals under similar conditions (Supplementary Table 5), which could only be obtained in the presence of maleate (Supplementary Figure 1d), an inhibitor that stabilizes the closed conformation when bound by both WT [Jäger J Mol Biol 1994] and HEX [Malashkevich Nat Struct Biol 1995].”

Supplemental Figures 8, 9, 10, and 15 need the associated PDB IDs for clarity.

Response: Thank you for the suggestion. We have added the associated PDB IDs in Supplementary Figures 8, 9, 10, and 15.

The Kirsch HEX study uses different residues numbers (V39L, K41Y, T47I, N69L, T109S, N297S). Stating “numbering based on Uniprot sequence P00509” is fine but at least give the reader the comparison, V35L, K37Y, T43I, T104S, and N285S. The last two are dismissed without discussion?

Response: To clarify the correspondence between numbering schemes, we have added the following sentences:

Table 1, footnotes: “Mutations are numbered based on Uniprot sequence P00509. V35, K37, T43 and N64 correspond to V39, K41, T47 and N69 in the previously published crystal structures of wild-type AAT (PDB ID: 1ARS and 1ART).”

Table 1, footnotes: “This variant also contains the T104S and N285S mutations located near the cofactor phosphate moiety (T109S and N297S according to the numbering from previously published HEX crystal structures [PDB ID: 1AHE and 1AHY]).”

Figure 2 caption: “Designed residues V35, K37, T43 and N64 correspond to V39, K41, T47 and N69 in the previously published crystal structures of wild-type AAT (PDB ID: 1ARS and 1ART).”

We did not discuss the T104S and N285S mutations from HEX because these positions were not included in the set of designed residues, and are therefore not present in any of our designed open or closed variants. Those mutations are however mentioned in the Table 1 footnotes (see above) when reporting the mutations found in HEX.

825 d This variant also contains the T104S and N285S active-site mutations. These are all active site mutants.

Response: To clarify, we have modified Table 1 footnotes as indicated below:

Table 1, footnotes: “This variant also contains the T104S and N285S ~~active-site~~ mutations located near the cofactor phosphate moiety”

The sequences in Supplement Table 10 should be a multiple-alignment with the N-termini indicated. Individual sequences are not very useful.

Response: Thank you for the suggestion. We have added a multiple sequence alignment as Supplementary Figure 16. Individual sequences are also provided for readers who may want to perform their own sequence alignments or order synthetic genes based on these sequences.

The PDB residue code for the LYS-PLP internal aldimine is LLP. This will help distinguish the structures in which PLP has become disconnected from K246. The malate substrate resides in the Michaelis site.

Response: In our crystal structures, we separated the Lys-PLP internal aldimine as two residues (PLP and LYS) to match the structures of WT *E. coli* AAT (PDB ID: 1ARS) and HEX (PDB ID: 1AHE, 1AHF, 1AHX, 1AHY) that we used in our analyses throughout this project. Our personal preference is to keep the internal aldimine separated into its constituents, as this facilitates comparison with the WT and HEX structures that informed our work. Furthermore, our structures have already been released and therefore redepositing modified coordinates to change the nomenclature could potentially introduce confusion for users of our models.

For simplicity, the structures should use the same crystallographic origin. VFIT apo, VFIY apo, HEX inhib, VFIT inhib, VFIY inhib, and VFCS inhib have different transform matrices. Not crucial but makes comparison easier.

Response: Our AAT structures can easily be aligned in Pymol or other modelling software for comparison. Furthermore, our structures have already been released and therefore redepositing modified coordinates to change the cell origin could potentially introduce confusion for users of our models.

REVIEWERS' COMMENTS

Reviewer #3 (Remarks to the Author):

The responses to the reviewer questions/concerns are adequate